# TOA: Task-oriented Active VQA

**Xiaoying Xing**    **Mingfu Liang**    **Ying Wu**
Northwestern University
Department of Electrical and Computer Engineering
{xiaoyingxing2026, mingfuliang2020}@u.northwestern.edu
yingwu@northwestern.edu

## Abstract

Knowledge-based visual question answering (VQA) requires external knowledge to answer the question about an image. Early methods explicitly retrieve knowledge from external knowledge bases, which often introduce noisy information. Recently large language models like GPT-3 have shown encouraging performance as implicit knowledge source and revealed planning abilities. However, current large language models can not effectively understand image inputs, thus it remains an open problem to extract the image information and input to large language models. Prior works have used image captioning and object descriptions to represent the image. However, they may either drop the essential visual information to answer the question correctly or involve irrelevant objects to the task-of-interest. To address this problem, we propose to let large language models make an initial hypothesis according to their knowledge, then actively collect the visual evidence required to verify the hypothesis. In this way, the model can attend to the essential visual information in a task-oriented manner. We leverage several vision modules from the perspectives of spatial attention (*i.e.*, Where to look) and attribute attention (*i.e.*, What to look), which is similar to human cognition. The experiments show that our proposed method outperforms the baselines on open-ended knowledge-based VQA datasets and presents clear reasoning procedure with better interpretability.

## 1 Introduction

The problem of knowledge-based visual question answer (VQA) [1, 2] requires open-world knowledge to answer the question about an image, which is more challenging than traditional VQA tasks [3] since the model needs to extract relevant external knowledge and then perform joint reasoning on the question, image, and the knowledge. To this end, most existing solutions explicitly retrieve external knowledge from various sources like Wikipedia [4, 5, 6, 7] and ConceptNet [8, 9, 4]. However, their performances are mainly limited by the explicit knowledge-retrieval stage [10]. First, the required knowledge may not be successfully retrieved, leading to insufficient information to answer the question. Second, the retrieved knowledge may be noisy and introduce irrelevant information, hence degrading the performance of the joint reasoning stage.

Recent works [11, 10, 12, 13] explore to use the large language models (LLMs) like GPT-3 [14] as implicit knowledge base, since they have shown powerful abilities in knowledge retrieval [15] and reasoning [16]. However, it remains a challenge to enable a pretrained LLM to further exploit the visual information, as this requires large computation resources to fine-tune the LLM on vision-language tasks [17, 18]. To sidestep this challenge, captioning-based methods translate the image into a caption such that the LLM can understand image information like a textual prompt. Despite the significant improvement compared to the explicit retrieval-based methods, their performances are still limited because the caption may usually drop essential visual information required to answer the question. On the other hand, the detection-based method leverages object detectors to extract all

the candidate visual concepts in the image and provide the LLM with textual object tags [5, 6] or brief description [12]. In this way, the LLM is provided with more exhaustive visual information. Nevertheless, many of the detected objects are irrelevant to the question and introduce noisy visual information to the language model. Another category of works utilizes the in-context learning ability of LLM to decompose the input questions into modular programs and execute on a set of pre-defined vision functions [19, 20]. However, they directly generate the answers through the execution of the program with simple symbolic logic, which limits their application to knowledge-based VQA.

To help the LLM better understand the image, we propose a task-oriented active (TOA) VQA method to actively and progressively acquire the visual information according to the task, as depicted in Figure 1. In general, our TOA method imitates the human cognitive process of reasoning, hypothesis, and verification. It consists of a LLM as the scheduler to make task planning and decision, and visual models to execute the order from the scheduler. Based on the understanding of the input question and the open-world knowledge obtained during pretraining, the LLM makes hypotheses on the answer and actively acquires relevant visual evidence from the visual model to verify the hypothesis. As a consequence, the LLM in our TOA method can integrate open-world knowledge and the collected visual evidence to answer the question, in contrast to previous works that directly obtain the answer only by a generated program [20, 19]. On the other hand, our method extracts the visual information in a task-oriented manner instead of looking at the whole image aimlessly, hence attending to the essential visual information more accurately and reducing the irrelevant information. Moreover, differing from the existing methods that generate a complete program at once and solely based on the question [20, 19], the LLM in our TOA interacts with the vision modality progressively through multi-round dialogue and decides the next instruction based on the previous interaction experience. When the visual evidence conflicts with the hypothesis, the LLM can make a new hypothesis or ask for other visual evidence to make the final decision. Therefore, even when the vision model makes a mistake, our method can still verify the hypothesis from a different angle.

To verify the effectiveness of the proposed method, we evaluate our methods and other competitive methods on OK-VQA [2] and A-OKVQA [2] datasets, where both datasets ask questions that require open-world knowledge beyond the image to obtain the correct answer. The empirical results show that our method outperforms the comparison methods on open-ended question answering, indicating our capability to better leverage implicit knowledge and capture essential visual information.

Our contributions are summarized as follows:

- We propose a new method to actively collect visual information in a task-oriented manner, which can more accurately attend to the essential information in the images and reduce the introduction of irrelevant information.

- We design a human-like cognitive process, *i.e.*, reasoning, hypothesis, and verification, to better activate the reasoning abilities of the LLM through clear problem-solving procedures and high-quality visual information.

- We develop a multi-round dialogue approach to solve the problem progressively and decide the next step dynamically, which has a clear answering process and better tolerance of mistakes in previous steps.

## 2 Related Work

**Knowledge-based Visual Question Answering.** Early works [21, 22, 23, 24] retrieve supporting knowledge from fixed knowledge bases annotated with the required facts to answer the questions. Subsequent open-knowledge VQA tasks [2, 1] need to acquire open-world knowledge instead of fixed knowledge bases. One category of solutions to open-knowledge VQA [4, 7, 8, 9] explicitly retrieves external knowledge from various sources, such as Wikipedia [25], ConceptNet [26] and Google Images [4]. The performances of the explicit retrieval-based methods are mainly limited by the knowledge retrieval stage, since it may either fail to retrieve the required knowledge or introduce noisy information. Another line of works [11, 10, 12, 13] leverages LLM as an implicit knowledge source. They often translate the visual information into captions [11, 13] or object-level descriptions [12]. Although the pioneering caption-based method PICa [11] significantly outperforms explicit retrieval-based methods, it is still limited since the generic caption may drop some essential visual information required to answer the question. To remedy this issue, PromptCAP [13] generates

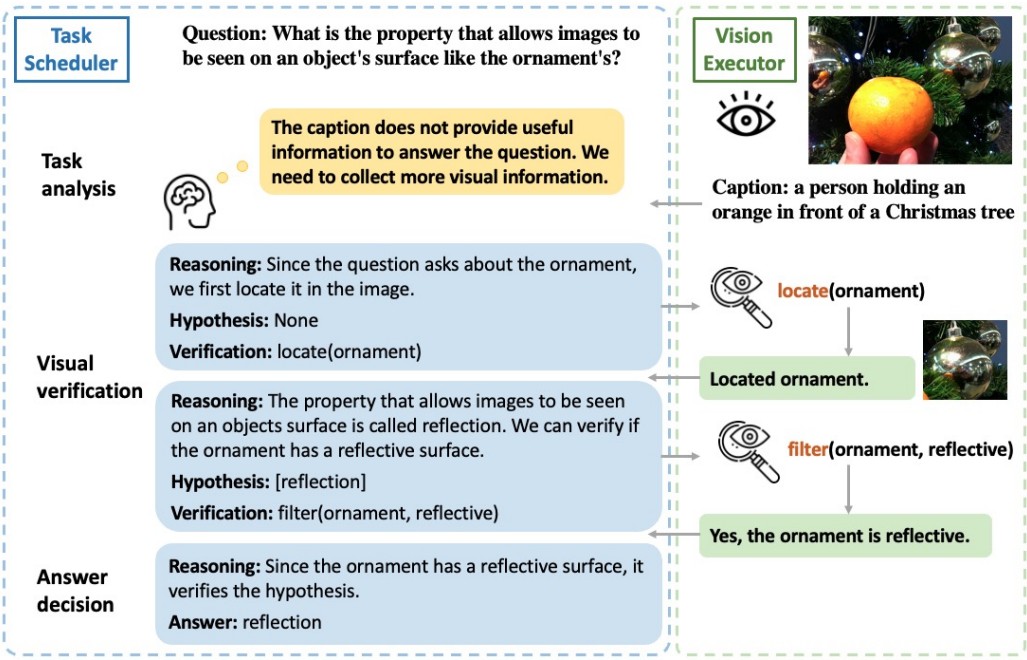

Figure 1: The overview of our task-oriented active (TOA) VQA method. TOA consists of a task scheduler and vision executor which communicates through multi-round dialogue. The task scheduler actively analyzes the existing information. Then it either requires specific visual information to help it make a further decision, or makes hypotheses on the answer and asks for visual evidence to verify its hypothesis. On the other hand, the vision executor follows the instruction from the scheduler to perform different kinds of visual verification and then returns the execution results to help the scheduler come up with the next decision. The gray arrows indicate the interaction process.

more relevant image captions conditioned on the input question. However, how to efficiently extract image information and translate it into textual prompts for LLMs remains an open problem. Different from the previous works that prompt LLMs with passively extracted visual information, we propose to let the LLM actively acquire the essential visual information in a task-oriented manner.

**Promping Large Language Model for Reasoning.** Recent works have found that LLMs emerge reasoning abilities [27, 16, 28, 29]. When prompted with in-context examples of chain-of-thought rationales, LLMs have shown impressive performance on tasks that require reasoning abilities [16, 29, 30]. Subsequent work [28] further explores the zero-shot reasoning ability of language models by simply prompting with 'Let's think step by step'. Another family of works excavate the task-planning ability of large language models [19, 20, 31, 32, 33]. They prompt the LLM to generate a programming over sub-tasks which are executed by pre-defined sub-modules. Among them Visual Programming [19] achieves impressive performance with good interpretability on GQA dataset [34], which is designed to examine the compositional reasoning ability. However, since the answer is generated through the program execution, it has limited application on open-knowledge VQA.

**Vision-language Model.** Vision-language pre-training models have made significant progress in recent years [35, 36, 37, 38, 39, 40]. They first pretrain the model on large-scale image-text data to learn a good image and language representation, then fine-tune on various down-streaming tasks. They exhibit strong transfer ability and achieve state-of-the-art on various vision-language tasks. Another line of works train the model by contrastive learning, aiming to align visual concepts and language representations [41, 42]. Besides, BLIP-2 [43] proposes a pre-training strategy to first learn vision-language representation learning from a frozen image encoder, then learn vision-to-language generative learning from a frozen language model. However, these vision-language pre-training models require large vision-language datasets and computation resource to fine-tune. Our method uses some pre-trained vision-language models to attend to the relevant region on the image and further collect the required visual information.

# 3 Method

In this section, we introduce our task-oriented active (TOA) VQA which simulates the cognitive process of humans involving three essential parts: reasoning, hypothesis, and verification. To this end, we design two agents, the task scheduler and the visual executor. These two agents engage in multi-round dialogues to communicate with each other. The scheduler parses the question, analyzes the required visual information, makes proper hypotheses to the visual executor for verification, and decides on the final answer with comprehensive consideration. The visual executor consists of a set of vision functions and collects visual information according to the requests from the scheduler. The illustration of the proposed method is shown in Figure 1.

## 3.1 Task Scheduler

The task scheduler leverages the natural language understanding and chain-of-thought [16] reasoning abilities of LLM, which plays the role of task analysis, planning, and decision-making. Given the input question and a brief description of the image generated by a captioning model [13], the LLM first parses the question and the information provided by the caption. Then it either makes an initial hypothesis and collects visual evidence to verify the hypothesis, or acquires more visual information when there is insufficient information to make any hypothesis. In the subsequent steps, the LLM makes further analysis considering the collected visual information. It can adjust the current hypothesis according to the previous conversation and ask for further verification. When the hypothesis is confirmed by the visual evidence, it decides the final answer and ends the dialogue.

Since the language model generates natural language based on a probabilistic model, they tend to generate diverse contents. To facilitate the interaction between the language modality and the vision models, we restrict the output format to a combination of free-form natural language and symbolic representation. Specifically, the output format consists of four parts: REASONING, HYPOTHESIS, VERIFICATION, ANSWER as the following template:

REASONING: $r$ HYPOTHESIS: $[h_1, h_2, ...]$ VERIFICATION: $function(args)$ ANSWER: $a$

At each step, the instructions generated by the scheduler contain all four parts above to guarantee format consistency. Each of them can be **None** when inapplicable. REASONING is the analysis basis of the subsequent parts. It contains a chain-of-though reasoning process that illustrates the hypothesis and requires visual verification. HYPOTHESIS is a list of possible answers to the question. There can be multiple hypotheses at each time, while when the current information is insufficient to make any hypothesis, then HYPOTHESIS would be marked as **None**. VERIFICATION specifies the instruction to call the vision functions. The instruction is constructed in Python similar to [20, 19], which consists of the name of the function and the specified args. ANSWER is the prediction result to the question. It is only specified when the scheduler has decided on the final answer, otherwise marked as **None**.

## 3.2 Visual Executor

Despite the strong reasoning and language understanding abilities of LLMs, they have limited capability for image understanding as fine-tuning on vision-language data requires large computation resource. To alleviate this issue, we leverage several vision models to execute the requests from the scheduler and collect the required visual evidence to verify the hypothesis. By doing so, the LLM can actively acquiring specific visual information that it believes to be crucial for enriching its understanding of the image to make an appropriate decision. Moreover, our proposed method extracts the visual information in a task-oriented manner, *i.e.*, tailored to the question, which can more efficiently attend to the useful information and reduce the noisy information than existing methods.

Motivated by human cognition, we correspondingly design our visual executor to have the task-oriented visual understanding module from two aspects, spatial attention (*i.e.*, Where to look) and attribute attention (*i.e.*, What to look). Instead of looking at the whole image aimlessly, our task-oriented image understanding module first locates the region of interest that is most relevant to the question. Then it extracts different image textures according to the query, including attribute-level features and semantics-level features. The functions of the vision executor are shown in Figure 2. We leverage several pre-trained vision-language models to implement the visual executor module, which will be elaborated in Section 4.1.

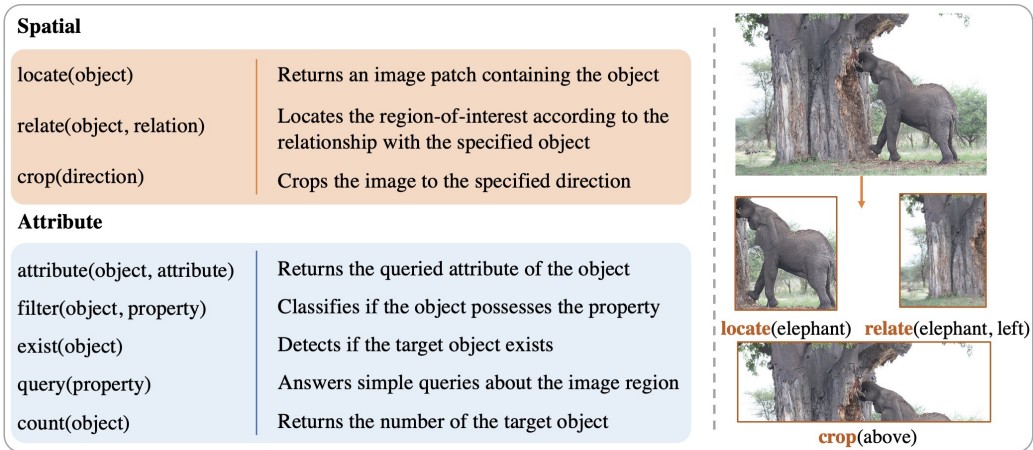

| Spatial | |
|---|---|
| locate(object) | Returns an image patch containing the object |
| relate(object, relation) | Locates the region-of-interest according to the relationship with the specified object |
| crop(direction) | Crops the image to the specified direction |

| Attribute | |
|---|---|
| attribute(object, attribute) | Returns the queried attribute of the object |
| filter(object, property) | Classifies if the object possesses the property |
| exist(object) | Detects if the target object exists |
| query(property) | Answers simple queries about the image region |
| count(object) | Returns the number of the target object |

Figure 2: Description of the spatial and attribute vision modules (left) and execution examples of spatial function (right).

## 3.3 Multi-rounds Interaction

In our proposed method, the task scheduler and the visual executor can interact with each other through multi-round dialogue. In this way, the scheduler in our method can acquire the essential information progressively, and it can also dynamically decide the next step based on the previous conversation. When there are multiple hypotheses that can be the possible answers to the question, the scheduler will verify each of them sequentially. If the visual verification conflicts with one of the hypotheses, then this hypothesis is excluded from all the hypotheses. However, sometimes all the hypotheses may be excluded by the visual verification. This may be caused by misleading results of the vision model or because all the hypotheses made by the scheduler are wrong. In such cases, the scheduler needs to jointly consider the common sense knowledge and the collected visual information. Thus we let the scheduler consider all the information at hand to either ask for more visual information to make a new hypothesis, or maintain the initial hypothesis and give the answer with the highest probability. Compared to the methods that directly generate the answer in the program execution [19, 20] and fully rely on the results of the vision models, our proposed method encourages the LLM to actively seek essential visual evidence oriented to the question in a sequential hypothesis-verification procedure. This may reduce the impact of the inferior results of vision models especially when asked to perform challenging vision tasks, since the LLM can make an alternative request which is more feasible to vision models.

When the scheduler confirms the answer through visual verification, it stops the dialogue and outputs the answer. Since the scheduler decides the cessation of the dialogue accordingly, in contrast to the previous works that end the dialogue after fixed rounds [44], our hypothesis-verification process through the dialogue can be more efficient and reduces redundant interactions.

## 3.4 Prompting Strategy

Now we introduce our prompting strategy to activate the reasoning ability of LLM. It consists of a brief description of the task requirements, instructions to leverage the available vision models, and a few in-context examples to enforce the LLM to generate the output in a consistent format. Specifically, the task description emphasizes the idea of hypothesis-verification, and prompts the task scheduler to leverage the vision functions to gather information. It is designed as follows:

> *I have a question about an image. Each time you need to make a hypothesis and verify by collecting visual information. You can use the following visual functions to gather information. You can only call one vision function at a time. The answer should be very concise.*

The instruction for leveraging the available vision models is Python-like pseudo code that describes the vision function, inspired by the API design in ViperGPT [20]. In this way, the scheduler can

better understand the function of the vision models, and it facilitates the scheduler to call the required vision functions. An example is shown as follows:

```
def filter(_object:str, _property:str)->bool:
    '''
    presupposes the existence of _object.
    '''
    return True if _object possesses the _property else False.
```

We also provide the large language model with a few in-context examples of the multi-round dialogue to guarantee the format consistency of the textual output. Previous works [45, 11] have indicated that in-context example selection is essential to the result. To find the most representative examples, we extract the image and question embedding using CLIP model [41]. Then we cluster the image-question pairs with the feature embedding and select the examples that are closest to the cluster centers as the representative examples. In the inference stage, we follow the in-context example selection approach in [11]. For each input data, we compute its cosine similarities with the available examples in the embedding space and select the top $k$ examples with the highest similarities. Specifically, for the training data $X = \{v_i, q_i\}_n$, we denote the fused image and question embeddings as $z_i = \left(v_{emb}^i, q_{emb}^i\right)$, and the clustering centers as $C = \{c_1, c_2, \cdots, c_m\}$. The representative examples of each cluster $c_j$ are selected as follows:

$$r_j = \arg\max_{z_i} \frac{c_j^\top z_i}{\|c_j\|_2 \|z_i\|_2}. \tag{1}$$

Then given a testing input image-question pair $(v, q)$ and $z = (v_{emb}, q_{emb})$, the top $k$ in-context examples with maximum similarities are:

$$\mathcal{I} = \underset{i \in \{1,2,\ldots,m\}}{\arg\text{TopK}} \frac{z^\top r_i}{\|z\|_2 \|r_i\|_2}. \tag{2}$$

## 4 Experiment

### 4.1 Experiment Settings

**Datasets.** We mainly evaluate our proposed method on knowledge-based VQA dataset OK-VQA [1] and conduct several experiments on A-OKVQA [2] as supplementary. OK-VQA is a commonly used knowledge-based VQA dataset that contains 14,055 image-question pairs associated with 14,031 images from MSCOCO dataset [46]. The questions are manually filtered to ensure all questions require outside knowledge to answer. Each question is annotated with ten open-ended answers. We evaluate our method on the test set. A-OKVQA is an augmented knowledge-related dataset containing 25K image-question pairs. It also provides supporting facts and rationales to obtain the answer. We conduct our experiments on the validation set. A-OKVQA benchmark encompasses both multiple-choice settings and direct-answer settings without answer options.

**Implementation Details.** In the experiments, we implement the task scheduler using gpt-3.5-turbo[1]. For the vision executor, we implement the spatial functions using Grounding DINO [47], which is an open-set object detector. The attribute functions are implemented by vision-language pre-training models BLIP2 [43] and X-VLM [48]. The representative in-context examples are selected using agglomerative clustering [49] with cosine metrics, and the feature embeddings are extracted by CLIP [41]. We prompt the scheduler with 16 in-context examples in the experiment.

**Evaluation Metrics.** The commonly used evaluation metrics for both OK-VQA and A-OKVQA direct-answer settings are the soft accuracy proposed in VQAv2 [3], where accuracy $= \min{(\#\text{humans that provided that answer}/3, 1)}$. The predicted answer is deemed 100% accurate if at least 3 humans provided the exact answer. However, with the rapid development of probabilistic generative models, the conventional exact matching evaluation can not meet the requirements of open-ended question answering. Since our answer is directly predicted by LLM without prior knowledge of the answer vocabulary, it may generate answers that are semantically equivalent to the ground truth but use different expressions. Thus, we evaluate our results with a combination of

---

[1]https://platform.openai.com/docs/models/gpt-3-5

Table 1: Comparison to the state-of-the-art methods on OK-VQA dataset. †: the result is based on reimplementation.

| Method | Knowledge Source | Image Representation | Accuracy |
|---|---|---|---|
| *Methods with explicit knowledge sources* | | | |
| ConceptBERT[9] | ConceptNet | Feature | 33.7 |
| KRISP [51] | Wikipedia+ConceptNet | Feature | 38.9 |
| Vis-DPR [52] | Google Search | Feature | 39.2 |
| MAVEx [4] | Wikipedia+ConceptNet+Google Images | Feature | 39.4 |
| KAT [5] | Wikidata+GPT-3 | Caption+Tags+Feature | 54.4 |
| REVIVE [6] | Wikidata+GPT-3 | Caption+Feature | 58.0 |
| *Methods using only implicit knowledge* | | | |
| IPVR [12] | GPT-3 | Caption | 44.6 |
| PICa-Base [11] | GPT-3 | Caption | 43.4 |
| PICa-Full [11] | GPT-3 | Caption | 48.0 |
| PromptCAP† [13] | GPT-3 | Caption | 58.8 |
| TOA (ours) | ChatGPT | Caption+Visual evidence | 60.6 |

Table 2: The results on A-OKVQA dataset. DA refers to direct answer settings and MC refers to multiple choices settings.

| Method | DA | MC |
|---|---|---|
| KRSIP [51] | 33.7 | 51.9 |
| IPVR [12] | 46.4 | - |
| PromptCAP [13] | 56.3 | 73.2 |
| TOA (ours) | 61.2 | 63.1 |

Table 3: Ablation study on important components of the proposed method.

| Model | Accuracy |
|---|---|
| TOA-full | 60.6 |
| TOA (w/o hypothesis) | 52.5 |
| TOA (w/o multi-round) | 57.8 |
| Visual Programming | 40.9 |

keyword matching and nearest-neighbor projection. Specifically, we first match the keywords of the predicted answer with the answer candidates. For the synonym paraphrasing without overlap of keywords, we project the prediction into the vocabulary using cosine similarity in the space of GloVe [50] word embeddings, which is similar to the multiple-choice evaluation of A-OKVQA. After this post-processing, we follow the conventional soft accuracy to report our results. It is worthwhile to note that the evaluation of open-ended answers is still an open problem and we believe that more efforts should be dedicated to addressing this problem given its increasing demand in the future.

## 4.2 Results on OK-VQA and A-OKVQA

We compare our proposed method with the state-of-the-art methods on OK-VQA dataset in Table 1. The compared methods are categorized into two classes. Methods in the first category [9, 51, 52, 4, 5, 6, 10] leverage explicit knowledge sources other than the implicit knowledge of LLM. The commonly used external knowledge source include ConceptNet [9, 51, 4], Wikipeida [51, 4, 5, 6] and Google [4, 52]. Note that KAT [5] and REVIVE [6] use both explicit knowledge sources and LLM as implicit knowledge. Methods in the second category [11, 13] only use LLM as the implicit knowledge source to obtain the answer. The results show that the introduction of LLM as an implicit knowledge source brings significant improvement compared to explicit knowledge retrieval methods. PromptCap [13] trains a task-aware captioning model with synthesized data. It is the primary baseline of our proposed method since the caption given to our task scheduler at the initial step is generated by PromptCap. The improvement in performance on our method indicates that we can acquire more useful visual information other than the caption through multi-round dialogue.

We also report the results on the A-OKVQA dataset in Table 2, where we outperform the compared methods on direct answer settings while achieving the second-best result on multiple choices settings. This implies that our proposed method is better at open-ended questions than multiple choices questions, since it generates open-ended answers of free-form natural language.

## 4.3 Ablation Study

We conduct extensive ablation studies to have an in-depth view of each component of our proposed method. Due to the expensive price of calling the OpenAI API and the limited visit frequency allowed to the public, we conducted our ablation studies on a randomly selected subset of OK-VQA.

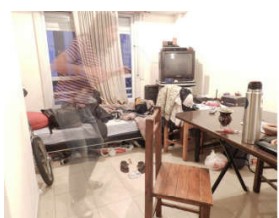

**Question:** Which furnishing seen here is of a type reminiscent of what is seen in some classrooms?

**Caption: a person walking in a messy room.**

**Reasoning:** Since the captioning does not provide any information about the furnishings, we need to locate them first.
**Hypothesis:** None **Verification:** locate(furnishing) **Answer:** None

**Response:** Located furnishing.

**Reasoning:** We can use visual attributes to compare the furnishings to those seen in classrooms. Desks and chairs are often seen in classrooms. We can verify if there exists a desk or chair.
**Hypothesis:** [desk, chair] **Verification:** exist(desk) **Answer:** None

**Response:** True

**Reasoning:** Since the vision model verifies the hypothesis, we can decide the answer.
**Hypothesis:** None **Verification:** None **Answer:** desk

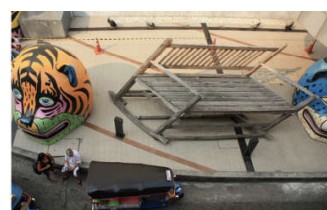

**Question:** What media was used to create the tigers face?

**Caption: a group of people sitting on a sidewalk next to two tigers.**

**Reasoning:** Since the question asks about the tigers face, we first locate it in the image.
**Hypothesis:** None **Verification:** locate(tiger face) **Answer:** None

**Response:** Located tiger face.

**Reasoning:** The media used to create the tigers face could be paint, pencil, or digital. We can verify if the tigers face is painted.
**Hypothesis:** [paint] **Verification:** filter(tiger face, painted) **Answer:** None

**Response:** True

**Reasoning:** Since the verification agrees with the hypothesis, we can decide the answer.
**Hypothesis:** None **Verification:** None **Answer:** paint

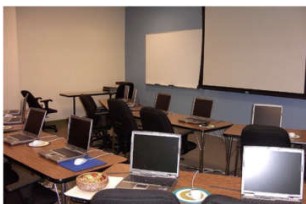

**Question:** What is the white screen on the back for?

**Caption: a classroom with laptops on the desks and a white screen on the back.**

**Reasoning:** We can infer from the captioning that the white screen is a projection screen, We can verify if there exists a projector in the image. **Hypothesis:** [projection] **Verification:** exist(projector) **Answer:** None

**Response:** False

**Reasoning:** Since there is no projector in the image, we assume that the white screen is used for writing or drawing with markers.
**Hypothesis:** [writing, drawing]
**Verification:** filter(white screen writing) **Answer:** None

**Response:** False

**Reasoning:** Since the verification conflicts with the hypothesis, we call the simple query function to obtain the answer.
**Hypothesis:** None **Verification:** simple query(What is the white screen on the back for?) **Answer:** None

**Response:** projector

**Reasoning:** The answer obtained from the simple query function is projection screen, which implies that the white screen is used for projecting images or videos from a projector.
**Hypothesis:** None **Verification:** None **Answer:** projector

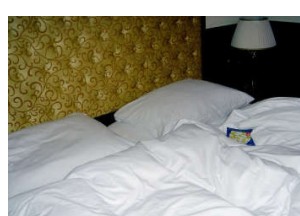

**Question:** What design is on the headboard of the bed?

**Caption: a bed with a design on the headboard.**

**Reasoning:** Since the question asks about the headboard of the bed, we first locate it in the image.
**Hypothesis:** None **Verification:** locate(headboard) **Answer:** None

**Response:** Located headboard.

**Reasoning:** We can use vision model to detect the design on the headboard. **Hypothesis:** None
**Verification:** query_attribute(headboard, design) **Answer:** None

**Response:** It is a gold patterned headboard.

**Reasoning:** Since the result shows that the headboard has a pattern, we can further query about the pattern. **Hypothesis:** [gold pattern]
**Verification:** query_attribute(headboard, pattern) **Answer:** None

**Response:** a floral pattern.

**Reasoning:** The vision model specifies the pattern on the headboard, so we can decide the answer. **Hypothesis:** None
**Verification:** None **Answer:** flower

Figure 3: Examples of multi-round dialogue between task scheduler and vision executor.

Table 4: Results of OK-VQA dataset with a different number of in-context examples and selection strategy.

|  | 8 examples | 16 examples | 24 examples |
|---|---|---|---|
| Random | 54.9 | 58.8 | 60.2 |
| Similarity-based | 55.3 | 60.6 | 60.6 |

There are two important components in our proposed method, *i.e.*, the multi-round dialogue strategy, and the hypothesis-verification process. To quantify the benefit of these two main strategies, we remove each of them from our full model and compare the results. To evaluate the impact of the hypothesis-verification strategy, we compare an ablative model which removes the REASONING and HYPOTHESIS parts from the instruction of the task scheduler. At each step, the scheduler directly specifies the vision function without explicit reasoning. To evaluate the benefit of multi-round dialogue, we let the task scheduler generate the whole program according to the initial hypothesis. We also compare the results of Visual Programming [19] since it is similar to our proposed method but generates the whole programming at one time. The results are shown in Table 3, and they demonstrate that the hypothesis-verification process with multi-round dialogue is essential to the performance. The clear output format based on the idea of hypothesis-verification better activates the reasoning ability of large language models. Multi-round dialogue allows the model to decide the next step dynamically based on the previous results, hence having better efficiency of visual information collection and tolerance of the misleading results of the vision model.

Another factor that influences the performance is the in-context example selection. Previous works [45, 11] have discussed the influence of in-context example selection strategies and the number of examples. We show the results of random selection and the heuristic selection described in Section 3.4, each with respect to a different number of examples in Table 4. We can observe from the results that more in-context examples often bring better results, while the marginal impact is diminishing. The similarity-based in-context selection strategy can achieve better results than random selection since it can pair the input question with the most similar examples. However, as the number of examples increases, the advantage is not as significant. This is probably because the model already has enough similar examples selected from the fixed available example pool.

## 4.4 Qualitative Results

We show a few qualitative results in Figure 3 to further illustrate our proposed task-oriented active VQA approach. In the upper left example, the captioning does not provide relevant information to answer the question. For the upper right example, the captioning fails to distinguish between real tigers and tiger-shaped statues. However, the scheduler makes a reasonable hypothesis based on common-sense knowledge, and the hypothesis is verified by the vision model. For the bottom left example, the vision model fails to detect the projection screen, and the responses conflict with the initial hypothesis. In this scenario, the scheduler makes a new reasonable hypothesis and actively acquires more information. Despite the failure of the vision model, it makes the correct decision considering both the common-sense knowledge and the visual information. Finally, the bottom right example shows how the scheduler progressively acquires the essential visual information based on the previous dialogue to give a more accurate answer.

## 5 Conclusion

We propose task-oriented active VQA (TOA), which uses LLM as an implicit knowledge source and answers the question through a sequential hypothesis-verification process. In TOA, the LLM as the task scheduler actively acquires visual evidence from a set of vision models in a task-oriented manner, and hence can attend to the essential information in the image more accurately and efficiently. Our proposed method dynamically decides the next step through multi-round dialogue. It can better deal with situations when the vision models are imperfect, though still limited by the performance of them. A potential limitation of our method is that the design of the prompting examples is subjective, which may bring uncertainty to the reasoning process. Besides, addressing the evaluation of open-ended question answering remains an open problem for future works.

## Acknowledgments

This work was supported in part by National Science Foundation grant IIS-1815561 and IIS-2007613.

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

# Appendix

## A    Demonstration of our prompt instructions and in-context examples

We illustrate the prompt instructions and in-context examples we used in our method in Figure 4.

## B    Human evaluation and Using GPT for evaluation

Since the final answers are generated open-ended and unaware of the answer vocabulary in our proposed method, it is improper to evaluate the results by conventional exact matching. To handle the semantically equivalent answers, we proposed a method combining keyword matching and nearest-neighbor projection. To further analyze our evaluation strategy, we conduct human evaluation on the predictions of OKVQA dataset and also leverage ChatGPT to judge whether the predicted answer and the ground truth can be considered consistent, given the original question. The results of different evaluation strategies are shown in Table 5.

These results further demonstrate that although our initial attempt on open-ended answer evaluation may not be perfect, it did not over-claim the performance of our proposed method. Thus we can affirm the effectiveness of our method. Moreover, this analysis further validates the necessity of addressing the evaluation of open-ended question answering.

Table 5: Comparison of results on OKVQA dataset by different evaluation methods.

|  | Matching and projection | Human | ChatGPT |
|---|---|---|---|
| Accuracy | 60.6 | 65.0 | 64.7 |

## C    Ablation study on the design choice of our prompting strategy

To demonstrate the effectiveness of our design principles, we conduct ablation study on different components of the prompting. We removed the task description, vision tools description and hypothesis guidance in the prompt instruction and in-context examples, respectively. The results are shown in Table C

The results show that these concise yet essential design principles have proven to be effective to guide the LLM for knowledge-based visual reasoning. However, we also recognize that our design might not be optimal. Further refinement in prompt design might yield improvements, which will be explored in our future work.

Table 6: Ablation study on prompting strategy.

| Model | Accuracy |
|---|---|
| TOA-full | 60.6 |
| w/o task description | 57.2 |
| w/o vision tools description | 58.6 |
| w/o hypothesis guidance | 55.1 |

**Prompting Instructions**

| | |
|---|---|
| **Task description** | I have a question about an image. Each time you need to make a hypothesis and verify by collecting visual information. You can use the following visual functions to gather information. |
| **Constraints** | You can only call one vision function at a time. The answer should be very concise. |
| **Vision tools description** | exist(object_name:str)->bool
        Returns True if the object specified by object_name is found in the image, and False otherwise.
attribute(object_name:str, visual_attribute:str)->str
        Returns the attribute of the object specified by object_name. Presupposes the existence of the object. The parameter visual_attribute is a nominal class name, such as color, shape, material.
filter(object_name:str, property:str)->bool
        Returns True if the object possesses the property, and False otherwise. Presupposes the existence of the object.
· · · |

**In-context examples**

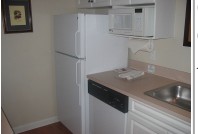

Question: Which of these appliances would you heat your noodles in?
Caption:  A kitchen with a refrigerator and a dishwasher
Assistant: **Reasoning**: We know that there is a refrigerator and dishwasher in the kitchen. None of them can heat. We often use microwave for heating. We can verify if there exists a microwave. **Hypothesis**: [microwave] **Verification**: exist(microwave) **Answer**: None
User:      True
Assistant: **Reasoning**: Since the hypothesis is verified by the vision model, we can decide the answer. **Hypothesis**: None **Verification**: None **Answer**: microwave

...

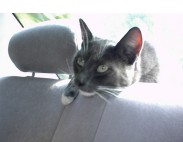

Question: Name the material used to make this car seat shown in this picture?
Caption:  A cat sitting on the back of a car seat
Assistant: **Reasoning**: Since the question asks about the car seat, we first locate it in the image. **Hypothesis**: None **Verification**: locate(car seat) **Answer**: None
User:      Located car seat
Assistant: **Reasoning**: Car seats are often made of leather, fabric or vinyl. We can directly verify each of them. **Hypothesis**: [leather, fabric, vinyl] **Verification**: filter(car seat, leather) **Answer**: None
User:      False
Assistant: **Reasoning**: The car seat is not made of leather. Next we verify if it is fabric. **Hypothesis**: [fabric, vinyl] **Verification**: filter(car seat, fabric) **Answer**: None
User:      True
Assistant: **Reasoning**: Since the vision model verifies the hypothesis, the answer is fabric. **Hypothesis**: None **Verification**: None **Answer**: fabric

Figure 4: The prompting instructions and in-context examples which guide the large language model to follow the hypothesis-verification reasoning stream. The instructions consist of task description, format constraints and vision tools description.

