# OpenReview forum: "TOA: Task-oriented Active VQA"
_NeurIPS.cc/2023/Conference — NeurIPS 2023 poster_

### Official Review · Reviewer_8uLs · 2023-07-04

**Soundness:** 3 good
**Presentation:** 3 good
**Contribution:** 3 good
**Rating:** 5
**Confidence:** 5

**Summary:**

This paper proposes task-oriented active VQA (TOA), which uses LLM as an implicit knowledge source and answers the question through a sequential hypothesis-verification process. This method can more accurately attend to the essential information in the images and reduce the introduction of irrelevant information. And they develop a multi-round dialogue approach to solve the problem progressively and decide the next step dynamically, which has a clear answering process and better tolerance of mistakes in previous steps. The experiments show the method outperforms the baselines and presents clear reasoning procedures.

**Strengths:**

1. The idea of this paper is very novel, and the key visual content is obtained through multiple rounds of dialogue.

2. The method takes full advantage of the rich knowledge and application flexibility of LLM.

3. The process design of reasoning-hypothesis-verification can reflect clear reasoning procedures and has better interpretability.

4. Experiments show that the performance of the method is very good, and verify the validity of the design idea.


**Weaknesses:**

1. Most of the sub-modules are somehow similar to existing models, and the design of the reasoning-verification process may be ad hoc and has great limitations.

2. In the  experiment, other methods use GPT-3 and this method uses ChatGPT, which is a bit unfair.


**Questions:**

Will this method be limited by the high cost of ChatGPT? What is the impact on the generation effect if other open-source LLMs are used?

**Limitations:**

The authors say that subjective example design brings uncertainty to the reasoning process, and evaluation methods for open-ended question answering are incomplete. I think it may be necessary to introduce some human evaluation as support in both aspects.

---

> ### Author Rebuttal · Authors · 2023-08-09
>
> Thank you for taking the time to review our manuscript, for providing such valuable feedback and your appreciation of our novelty regarding the knowledge-based VQA and LLM, the better interpretability of our approach, and our experimental results! Here are our response for each of your questions.
>
> **[Reviewer 8uLs Weakness-1: Choices of vision models and paradigm design]**: We appreciate the reviewer's concern. First, we want to emphasize that: our core contribution lies in creating a novel computational paradigm where the LLM can iteratively and sequentially combine question and image information, imitating human cognitive processes and using its knowledge for reasoning. Although we utilize existing sub-modules, it is our unique integration and the underlying philosophy that sets our approach apart. This new approach allows the LLM to actively form and verify hypotheses by collecting visual information, representing a significant innovation in the field.
>
> - **Regarding using existing vision models in our vision executor**: We utilized common vision models (e.g., vision models used in Visual Programming) to ensure a **fair and meaningful comparison** with existing methods. This is because only by doing so can we readily show that the improvement brought by our method is not because we are using different or more advanced vision models, but from the novelty of our proposed task-oriented active VQA approach for Knowledge-based VQA. This can be seen from the comparison between our proposed method and the Visual Programming in our main paper Table 3. Moreover, our flexible prompting design can allow the vision modules to be easily added or changed for different task requirements in our proposed method.
>
> - **Regarding "the design of the reasoning-verification process"**: Our hypothesis-verification reasoning process is inspired by the human cognitive process, where assumptions and inferences are made based on their knowledge and then verified actively. Translating this into a computational form is non-trivial, and our work is the first to tackle this challenging problem. We acknowledge that our implementation can be enhanced, but we believe our current work has effectively demonstrated the potential and feasibility of this new paradigm.
>
> **[Reviewer 8uLs Weakness-2: Using GPT-3 for our approach]**: Thank you for highlighting this concern. We observe that both GPT-3 and ChatGPT possess their unique strengths, and it's not a matter of one definitively being better than the other. GPT-3 tends to have stronger imitation abilities through in-context learning, whereas ChatGPT may have better reasoning capabilities and creativity, especially in multi-round dialogues. However, this might also make maintaining format consistency across multiple dialog rounds more challenging with ChatGPT.
>
> Our method involves using a multi-round dialogue format, which initially made ChatGPT a more convenient choice for us. To address your concern, we conducted additional experiments by switching from ChatGPT to GPT-3, the same model used in our compared baselines. Interestingly, the results showed that our method performed even better with GPT-3 than with ChatGPT (improved from **60.6%** to **61.5%**). This more even-handed comparison further accentuates the advantages of our proposed method over compared methods. Our conjecture for this interesting outcome is that GPT-3 already has sufficient reasoning ability and knowledge for current knowledge-based VQA tasks, and it may better imitate the in-context examples and follow the instructions.
>
> **[Reviewer 8uLs Question-1: Usage of other LLMs]**: Thank you for this question. We would like to point out that all the methods we have compared, based on LLMs, are dependent on GPT. Thus, in general, all related methods would be somewhat constrained by the cost of reasoning with GPT. Compared to existing methods such as Visual Programming, our method has significantly reduced the average reasoning cost for a single question from approximately 0.049 to 0.0085 dollars.
>
> Regarding open-source LLMs, the current open-source models still have some gaps in reasoning ability compared to the GPT series, which may somewhat impact the final answer's effectiveness. We conducted experiments with LLaMa and the free-to-use Claude-2 model by Anthropic. The accuracy of OKVQA using LLaMa is **51.8%**, which is still at an acceptable level. We believe that as open-source LLMs continue to advance, this issue will gradually be resolved. Our experiments on Claude-2 yielded an accuracy of **60.1%**, which is comparable to GPT, further demonstrating the applicability of our framework to different LLMs.

---

> > ### Comment · Reviewer_8uLs · 2023-08-20
> > **Thanks for the response**
> >
> > My major concerns have been addressed. I will keep my score.

---

### Official Review · Reviewer_TKN8 · 2023-07-06

**Soundness:** 3 good
**Presentation:** 4 excellent
**Contribution:** 3 good
**Rating:** 7
**Confidence:** 4

**Summary:**

This paper addresses knowledge-based visual question answering by leveraging a large language model (LLM) as a knowledge source. To mitigate the limited capability of vision models, the paper suggests letting the LLM predict hypothesis and actively gather visual evidence. Experimental results show that the proposed method outperforms baseline approaches in open-ended knowledge-based VQA.

**Strengths:**

- The paper is well-written, clearly structured, and easy to follow.
- The motivation behind using an LLM for VQA to assist imperfect vision models is reasonable, and qualitative results demonstrate that the method behaves as intended.
- Ablation studies reveal that the multi-round dialogue strategy and hypothesis-verification process positively contribute to the approach.
- The experiments are thorough, offering insights into the impact of design choices.

**Weaknesses:**

- The method's reliance on heuristic prompting may limit its generalizability to other LLMs.
- The validity of post-processing, which paraphrases predictions using a predefined vocabulary, is unclear. A small analysis of the post-processing would make the paper more convincing. For instance, providing examples of raw predictions and post-processed ones would help readers better understand the post-processing.

**Questions:**

- About the design choice of heuristic prompting, investigating the effects of different prompts would be interesting.
- It would be helpful to see in-context examples for specific image-question pairs.
- A table reference error (Table 4.1) should be corrected.

**Limitations:**

The paper discusses two limitations.
- Evaluating open-ended VQA remains an unresolved challenge.
- The design of prompting examples may not be optimal.
They are important issues in this area.

---

> ### Author Rebuttal · Authors · 2023-08-10
>
> Thank you for your appreciation of our writing, motivation, and experimental results! We have considered all your suggestions and have made the necessary revisions or clarifications to improve the quality of our manuscript as follows. We have attached a one-page pdf in our global response to demonstrate our prompting instruction and in-context learning examples.
>
> **[Reviewer TKN8 Weakness-1: Prompting strategy]**: We would like to apologize if naming Section 3 as 'heuristic prompting' in our paper gave the reviewer the impression that the design of the instruction prompting template was heuristic or without basis. Actually, it is designed to guide LLM reasoning following a thoughtful and deliberate process that includes several key elements:
>
> 1. **Defining the Task**: We begin by defining the task to inform the LLM about the specific reasoning task at hand.
> 2. **Introducing Available Vision Tools**: We then present the available vision tools within the prompt, enabling the LLM to recognize the executable functions to verify the hypotheses.
> 3. **Specifying Output Format Constraints**: By delineating the expected output format, we ensure consistent parsing of the LLM's output to automatically query our designed vision functions.
>
> You can also check the one-page pdf in global response for more concrete examples of our prompting strategy.
>
> **Ablation study on the design choice of our prompting strategy.** To demonstrate the effectiveness of our design principles, we have added a new ablation study. We removed the task description, vision tools description and hypothesis guidance in the prompt instruction and in-context examples, respectively. The results are shown in the table below:
> | Model                        | Accuracy (%) |
> |------------------------------|--------------|
> | TOA-full                     | 60.6         |
> | w/o task description         | 57.2         |
> | w/o vision tools description | 58.6         |
> | w/o hypothesis guidance      | 55.1         |
>
> The results show that these concise yet essential design principles have proven to be effective to guide the LLM for knowledge-based visual reasoning. However, we also recognize that our design might not be optimal. Further refinement in prompt design might yield improvements, which will be explored in our future work.
>
> **Applicability to other LLMs.** Last but not least, we change the GPT to the free-to-use Claude-2 model by the Anthropic and conducted the same experiment on OKVQA, where we can achieve comparable accuracy **60.1%** to GPT. This further demonstrating the applicability of our framework to different LLMs.
>
>
> **[Reviewer TKN8 Weakness-2: Analysis of the evaluation process]**: Thank you very much for the suggestion. Here we provide some examples of raw predictions and mapping to the answer vocabulary:
>
> (1) Prediction: to advertise; Ground Truth: advertising
>
> (2) Prediction: to tell directions; Ground Truth: navigation
>
> (3) Prediction: one-way traffic; Ground Truth: one way
>
> (4) Prediction: bike; Ground Truth: bicycle
>
> From the above examples, we can see that these matchings are reasonable and semantically highly relevant.
>
> Moreover, to analyze our evaluation strategy more comprehensively, we have further conducted two different types of evaluation:
>
> - **Human Evaluation**: We conducted a human evaluation on the predictions of OKVQA dataset by our proposed method, and the accuracy is **65.0%**, which is even higher than the result of 60.6% obtained by our original evaluation strategy in Table 1 of our main paper.
>
> - **Leverage GPT for evaluation**: After we obtain the predicted results, we separately call the OpenAI API to initialize a new GPT to judge whether the predicted answer and the ground truth can be considered consistent, given the original question. The resulting accuracy is **64.7%** for the prediction from our proposed method, which is similar to the human evaluation.
>
> These results further demonstrate that although our initial attempt on open-ended answer evaluation may not be completely perfect, it **did not over-claim** the performance of our proposed method. Thus we can affirm the effectiveness of our method. Moreover, this analysis further validates the necessity of addressing the evaluation of open-ended question answering.
>
>  **[Reviewer TKN8 Question-1]**: Please refer to our reply to **[Reviewer TKN8 Weakness-1]** for the **Ablation study on the design choice of our prompting strategy**.
>
>  **[Reviewer TKN8 Question-2: Demonstration of in-context examples]**: Please refer to the one-page pdf we attached in our global response for the demonstration of the in-context examples.
>
>  **[Reviewer TKN8 Question-3: Typo]**: Thank you for pointing this typo out; we have revised it accordingly!

---

> > ### Comment · Reviewer_TKN8 · 2023-08-12
> > **Thanks for your response**
> >
> > Thank you for your responses.
> > - The explanation has helped clarify the intuition behind designing the prompt, and that the ablations effectively validate the design.
> > - I appreciate the additional report on the results with Claude-2. The report suggests the generalizability of the proposed method.
> > - The examples of post-processing outputs were able to address my question. I believe that the technique is reasonable.
> >
> > The responses clarify the contributions of this work, thus I remain positive on this paper.

---

### Official Review · Reviewer_ckp5 · 2023-07-07

**Soundness:** 4 excellent
**Presentation:** 4 excellent
**Contribution:** 4 excellent
**Rating:** 7
**Confidence:** 4

**Summary:**

Early methods for Knowledge-based visual question answering (VQA) explicitly retrieve knowledge from external knowledge bases, often introducing noisy information. Current large language models like GPT-3 as implicit knowledge sources cannot effectively understand image inputs. Thus, extracting the image information and inputting it into large language models remains an open problem. Using image captioning and object descriptions to represent the image may either drop the essential visual information to answer the question correctly or involve irrelevant objects to the task of interest. To address this problem, the authors propose to let large language models make an initial hypothesis according to their knowledge, then actively collect the visual evidence required to verify the hypothesis. In this way, the model can attend to the essential visual information in a task-oriented manner. The authors leverage several vision modules from the perspectives of spatial attention (i.e., Where to look) and attribute attention (i.e.,16 What to look), which is like human cognition.

**Strengths:**

- The idea to exploit ChatGPT to solve knowledge-based VQA interactively seems novel.
- Compared to the recent similar method, Visual Programming, the proposed method shows favorable performance, and the authors provided an ablation study and hyper-parameter analysis to verify the characteristics of each component of the proposed method.

**Weaknesses:**

- It would be more helpful to add an experiment to verify whether the proposed method can be generally applied to other tasks. For example, can the proposed ChatGPT-based interactive framework also help improve the performance of other datasets, such as the VQAv2 dataset?

**Questions:**

Please refer to the questions in the weakness.

**Limitations:**

I cannot find a potential negative societal impact in this paper.

---

> ### Author Rebuttal · Authors · 2023-08-09
>
> We are very grateful for your insightful comments, especially your appreciation of the novelty of our proposed approach and our experimental results. For the concern you expressed in the Weakness section, we have addressed it in our **Global Response-2**. We paste the response here for your convenience.
>
> **[Global Response-2: About experiments on more datasets]** We greatly value your suggestions regarding the expansion of our experiments to more datasets and different vision-language tasks. We acknowledge the importance of demonstrating generalizability, but here's why we have opted for a specific focus on Knowledge-based VQA:
>
> - **Our specific focus: Knowledge-based VQA**: First, as explicitly demonstrated in our abstract and introduction, our method targets exploiting LLM for the Knowledge-based VQA task. This is because Knowledge-based VQA requires open-world knowledge to answer the question about an image and is more challenging than traditional VQA tasks [3] since the model needs to extract relevant external knowledge and then perform joint reasoning on the question, image, and the knowledge. Such a challenging requirement essentially demands the LLM to be actively engaging in collecting the essential information from both the question and image, and motivates us to propose our task-oriented active VQA method for Knowledge-based VQA in this paper. Extending our method to more general vision-language tasks is not our current priority in this paper, because we believe that it is more beneficial and important for us to comprehensively demonstrate the effectiveness of our proposed method on the challenging Knowledge-based VQA.
>
> - **Existing datasets for Knowledge-based VQA**: For the task of knowledge-based VQA, OKVQA and A-OKVQA datasets are the standard benchmarks in the existing literature [9, 51, 52, 4, 5, 6, 12, 11, 13]. The comparison of these two datasets with state-of-the-art methods in Table 1 and 2 in our main paper sufficiently demonstrates the efficacy of our proposed method for the knowledge-based VQA.
>
> - **More results on other VQA task**: As complementary, we have conducted additional experiments on the VQAv2 dataset to demonstrate the broader applicability of our method.  We achieved **74.8%** (**16 shots**) on a randomly selected subset of VQAv2 val set, which is a competitive few-shot result, compared with the powerful vision-language model Flamingo[a] (**68.4%, 32 shots**). We consider these exploratory results as a stepping stone for future research.
>
> [a] Flamingo: a visual language model for few-shot learning. NeurIPS 2022.

---

> ### Comment · Reviewer_ckp5 · 2023-08-17
>
> The authors' response answered my question.
> Therefore, I will keep my score for this paper.

---

### Official Review · Reviewer_BgCz · 2023-07-09

**Soundness:** 2 fair
**Presentation:** 3 good
**Contribution:** 3 good
**Rating:** 5
**Confidence:** 5

**Summary:**

This paper tries to solve the knowledge-based visual question answering (VQA) by proposing a new approach that utilizes LLMs for calling visual modules in a task-oriented manner. The method employs a reasoning-hypothesis-verification process in multiple rounds to progressively find the answer. Evaluations are conducted on OK-VQA and A-OKVQA to demonstrate the effectiveness of this method.

**Strengths:**

1. Multi-round interactions and reasoning-hypothesis prompting are introduced in this paper. Compared with previous one-time program generation, those two improvements are reasonable intuitively and can benefit on reported benchmarks.

**Weaknesses:**

1. In the title, abstract and introduction, `task-oriented` seems to be emphasized as the main advantage of this paper over others. However, ViperGPT and Visual Programming are also task-oriented. The main difference between this paper and the previous two are rationale and multi-rounds. The corresponding part should be rewritten.
2. I don't understand the necessity of the hypothesis. From the given examples, it seems without an assumed hypothesis, the LLM should still be able to call the right verification function. In Fig3's examples, imagine you remove the hypothesis, the workflow of LLM seems still smooth and reasonable, and many hypotheses are actually None. In Tab3's ablation, both reasoning and hypothesis are removed which is not indicative enough. Can you do an ablation to remove the hypothesis while preserving reasoning and verification and answer?
3. Why don't you compare with ViperGPT in OK-VQA? Is that because ViperGPT is in a zero-shot manner, while yours is in a few-shot manner? Then why your method must be reported in a few-shot manner and what performance it can achieve in the same manner as ViperGPT?
4. Only two benchmarks are compared. In ViperGPT/Visual Programming, 4/3 datasets are benchmarked. To demonstrate the generalization ability of the proposed method, at least one or two more typical datasets are needed.
5. Similar ideas of multi-rounds and reasoning in each step have been used in [1]. Please compare the difference.
6. Many visual models are employed in this paper however the technical details as well as related prompts are not included in the paper. It makes reproduction difficult.

Ref:
[1] See, Think, Confirm: Interactive Prompting Between Vision and Language Models for Knowledge-based Visual Reasoning

**Questions:**

See weaknesses.

**Limitations:**

The authors didn't discuss limitations in the paper. However, it might happen that the LLM (used in paper) generates some toxic or biased response.

---

> ### Author Rebuttal · Authors · 2023-08-09
>
> We appreciate your valuable and constructive comments. We have made the necessary revisions or clarifications to improve the quality of our manuscript as follows. We have attached a one-page pdf in our global response to demonstrate our prompting instruction and in-context learning examples. Here are our responses to each of your questions.
>
> **[Reviewer BgCz Weakness-1: The term of `task-oriented`]**: Thank you for your suggestions! We want to clarify that the term `task-oriented` refers to a dynamic, iterative process where the Large Language Model (LLM) actively engages with both the question and image. The LLM formulates hypotheses and conducts active vision verifications, adaptively reasoning based on the results until the LLM itself determines the final answer. This whole process is intimately connected to the task at hand, continually interacting with and adapting to the information gathered, which justifies our use of the term `task-oriented`.
>
> In contrast, existing methods such as Visual Programming and ViperGPT use the LLM merely as a tool for task planning or code generation. They generate an immutable program based solely on the question **without deeply or actively** engaging with the image, and they obtain the answer directly from the execution of vision models. Thus they are **not** `task-oriented` under our definition. In these methods, the LLM can not make adjustments for errors in either program planning or vision models, and their application on questions requiring open-world knowledge is limited.
>
> We appreciate your concern, and we will revise the relevant sections to make this distinction clearer and underline the unique aspects of our method.
>
> **[Reviewer BgCz Weakness-2: Necessity of Hypothesis]**: Please refer to the **Global Response-3**.
>
> **[Reviewer BgCz Weakness-3: Zero-shot setting]**: Our approach and all other compared methods need to employ few-shot instructions. This is because the in-context examples enable the LLM to follow our expected sequential hypothesis-verification reasoning process and give the output as formatted. The output format facilitates the distillation of the essential information from free-form text generated by LLM into executable visual functions. In our proposed method, the LLM acts as the reasoning agent like a human brain which conducts reasoning and task planning by actively utilizing its common-sense knowledge and finally outputs the answer by the LLM itself. While for the ViperGPT, it uses CodeX to directly generate Python code based on the question and derive answers from the execution of the generated code. Since the CodeX is pretrained specifically for code generation, ViperGPT does not require additional instructions to guarantee the output format and can work in a zero-shot setting.
>
> Given these fundamental differences between the two types of methods, we compare our method with the methods that also employ the LLM as a reasoning agent in a few-shot manner in our experiment section. Such comparison is fair, and our experiments have sufficiently demonstrated the effectiveness of our method.
>
> **[Reviewer BgCz Weakness-4: More datasets]**: Please refer to the **Global Response-2**.
>
> **[Reviewer BgCz Weakness-5: Comparison to IPVR[12]]**: Thank you for bringing the IPVR work to our attention, where we have already cited it and discussed it as IPVR [12] in our main paper with explicit comparison in both **Table 1 and 2** in the main paper. In IPVR, the process consists of three key modules: a 'see' module for object-level detection, a 'think' module for selecting attended objects and transforming them into captions, and a 'confirm' module for rationale verification. While the superficial similarities exist, we would like to highlight the fundamental differences between IPVR and our method:
>
> 1. **Object-level detection vs. Active image understanding**: In IPVR, they focus on object-level detection and attending, which may either bring irrelevant objects to the question or drop essential information in the caption. However, our method is not constrained by object-level image understanding and captioning, as we utilize various vision functions to actively acquire visual information.
>
> 2. **Confirmation Process**: While IPVR also involves a confirmation process, they merely use the LLM to generate a **textual** rationale to support the answer. They confirm the answer when the generated rationale matches the predicted answer. Contrarily, our method's verification part allows the LLM to acquire **new visual evidence** by vision executor based on its hypothesis, and facilitates subsequent rounds of reasoning. The LLM confirms the answer when it has obtained sufficient information to reason about the answer using its open-world knowledge.
>
> By elucidating these differences, we establish that our method, although superficially similar to IPVR, is **essentially different** in its approach and underlying principles. These differences contribute to the uniqueness and innovation of our proposed method. Moreover, these underlying disparities are manifest in our performance results. Specifically, our method's scores of **60.6** and **61.2** on the OK-VQA and A-OKVQA benchmarks, respectively, are significantly superior to IPVR's scores of **44.6** and **46.4** on the same benchmarks, as shown in Tables 1 and 2 in our main paper.
>
> **[Reviewer BgCz Weakness-6: Technical details]**: In designing and implementing our vision executor, we considered commonly used vision models in line with the methods compared in Table 1 and 2 with public code. This makes experimental comparisons fairer, and the re-implementation will not be difficult, since all the models we used have user-friendly API to call. This design choice allows us to highlight the improvements and advantages of our new method over existing methods by eliminating differences in vision models. We will publicly release our full implementation after acceptance and offer help for reproduction.

---

> > ### Comment · Reviewer_BgCz · 2023-08-17
> > **Towards Response.**
> >
> > Thank the authors for the explanation and supplementary experiments. It's suggested to highlight the explanation about how this model is more `task-oriented` compared with others at the beginning of the paper. Other concerns of mine have been resolved. I'd like to raise the rating.

---

### Author Rebuttal · Authors · 2023-08-09

We sincerely thank all reviewers for your insightful and constructive reviews! In this global response, we want to address the common questions inquired by different reviewers, and also demonstrate the necessity of making hypotheses in our proposed method.

**[Global Response-1: Demonstration of our prompt instructions and in-context examples]** In the one-page pdf allowed by the rebuttal, we provide figures to illustrate the prompt instructions and in-context examples we used in our method. We hope this can help reviewers better comprehend the simple yet effective design principles of our prompts.

**[Global Response-2: About experiments on more datasets]** We greatly value the suggestions from Reviewers **BgCz** and **ckp5** regarding the expansion of our experiments to more datasets and different vision-language tasks. We acknowledge the importance of demonstrating generalizability, but here's why we have opted for a specific focus on Knowledge-based VQA:

- **Our specific focus: Knowledge-based VQA**: First, as explicitly demonstrated in our abstract and introduction, our method targets exploiting LLM for the Knowledge-based VQA task. This is because Knowledge-based VQA requires open-world knowledge to answer the question about an image and is more challenging than traditional VQA tasks [3] since the model needs to extract relevant external knowledge and then perform joint reasoning on the question, image, and the knowledge. Such a challenging requirement essentially demands the LLM to be actively engaging in collecting the essential information from both the question and image, and motivates us to propose our task-oriented active VQA method for Knowledge-based VQA in this paper. Extending our method to more general vision-language tasks is not our current priority in this paper, because we believe that it is more beneficial and important for us to comprehensively demonstrate the effectiveness of our proposed method on the challenging Knowledge-based VQA.

- **Existing datasets for Knowledge-based VQA**: For the task of knowledge-based VQA, OKVQA and A-OKVQA datasets are the standard benchmarks in the existing literature [9, 51, 52, 4, 5, 6, 12, 11, 13]. The comparison of these two datasets with state-of-the-art methods in Table 1 and 2 in our main paper sufficiently demonstrates the efficacy of our proposed method for the knowledge-based VQA.

- **More results on other VQA task**: As complementary, we have conducted additional experiments on the VQAv2 dataset to demonstrate the broader applicability of our method.  We achieved **74.8%** (**16 shots**) on a randomly selected subset of VQAv2 val set, which is a competitive few-shot result, compared with the powerful vision-language model Flamingo[a] (**68.4%, 32 shots**). We consider these exploratory results as a stepping stone for future research.

**[Global Response-3: the necessity of the hypothesis-making]** We thank reviewer **BgCz** for inquiring about this question. We believe it is better to treat it as a global response to provide a comprehensive answer. Here is our response:

- **Necessity of Hypothesis**: In our proposed method, the hypothesis is derived from the knowledge of LLM and the current information that it processes. With a clear hypothesis-verification process, LLM will be prompted to actively incorporate and exploit its open-world knowledge for its reasoning, instead of completely relying on the output of the vision models. For instance, when the output of the vision models contradicts common-sense knowledge from the LLM, the LLM instructed by our hypothesis-verification mechanism will tend to collect more information from the image, e.g., make another hypothesis or change to a different verification tool, such that the LLM itself can integrate more comprehensive information to give the answer. Such merit has also been demonstrated in our main paper’s **lines 299-303**, with an example in the bottom-left subfigure of **Figure 3**.

- **The reason why the workflow is smooth in Figure 3**: When we design our instruction and in-context examples, we explicitly instruct the LLM to make hypotheses and verify them in the 'Reasoning' section in our prompting. **Thus the reasoning process of LLM has intrinsically included the process of hypothesis-making, and they are interconnected.** We distinguish ‘Hypothesis’ to keep the output format consistent and clear.

- **Regarding 'many hypotheses are actually None’**: This is due to the multi-round dialogue nature of our demonstration in Figure 3. It appears that many hypotheses in Figure 3 are ‘None’ in single steps, but that does not mean the model did not formulate them in the whole dialogue. In every single round, the ‘Hypothesis’ may be ‘None’ in three cases:
1. LLM lacks sufficient information to make any hypothesis and needs to gather more visual information, often at the beginning of the dialogue.
2. The previous hypothesis is overturned, and a new hypothesis is not made.
3. The answer can be decided by the collected information, and no more hypotheses are required. This is the final round of the dialogue, and the ending is determined adaptively by the LLM agent.

- **Ablation of removing the hypothesis**: Finally, we conducted two supplementary experiments. We first removed only the ‘Hypothesis’ from the output format. It does not obviously influence the overall accuracy, since the LLM is still following the hypothesis-verification stream. Then we rewrote the ‘Reasoning’ in our prompts to **remove the guidance for LLM to make hypotheses** during the ‘Reasoning’, and the performance on the OKVQA dataset dropped significantly from **60.6%** to **55.12%**, which emphasizes the **indispensability of the hypothesis** in our proposed method.

[a] Flamingo: a visual language model for few-shot learning. NeurIPS 2022.

---

### Decision · Program_Chairs · 2023-09-21

**Decision:**

Accept (poster)

**Comment:**

All reviewers are quite positive about this submission and no negative scores are given. All reviewers confirmed the contributions and novelty of this submission. And the authors' rebuttal addressed the reviewers' concerns very well. After the discussion and final update from reviewers, we all voted to accept this submission.